# Study on the Role of Salicylic Acid in Watermelon-Resistant Fusarium Wilt under Different Growth Conditions

**DOI:** 10.3390/plants11030293

**Published:** 2022-01-22

**Authors:** Feiying Zhu, Zhiwei Wang, Wenjun Su, Jianhua Tong, Yong Fang, Zhengliang Luo, Fan Yuan, Jing Xiang, Xi Chen, Ruozhong Wang

**Affiliations:** 1Hunan Provincial Key Laboratory of Phytohormones, College of Bioscience and Biotechnology, Hunan Agricultural University, Changsha 410128, China; feiyingzhu@hunaas.cn (F.Z.); tongjh0421@hunau.edu.cn (J.T.); elijash@139.com (F.Y.); XJ459970488@163.com (J.X.); 2Hunan Agricultural Biotechnology Research Institute, Hunan Academy of Agricultural Sciences, Changsha 410125, China; yongfang@hunaas.cn (Y.F.); luozl@hunaas.cn (Z.L.); xichen@hunaas.cn (X.C.); 3Hunan Agricultural Equipment Research Institute, Hunan Academy of Agricultural Sciences, Changsha 410125, China; wangzhiwei119@hunaas.cn; 4Zhuzhou Institute of Agricultural Sciences, Zhuzhou 412007, China; suwenjunw@163.com

**Keywords:** salicylic acid, watermelon, Fusarium wilt, resistance

## Abstract

Background: Fusarium wilt disease is leading threat to watermelon yield and quality. Different cultivation cropping systems have been reported as safe and efficient methods to control watermelon Fusarium wilt. However, the role of salicylic acid (SA) in watermelon resistance to Fusarium wilt in these different cultivation systems remains unknown. Methods: in this experiment, we used RNA-seq and qRT-PCR to study the effect of SA biosynthesis on improving watermelon health, demonstrating how it may be responsible for Fusarium wilt resistance under continuous monocropping and oilseed rape rotation systems. Results: the results revealed that the expression of the CIPALs genes was key to SA accumulation in watermelon roots. We observed that the NPR family genes may play different roles in responding to the SA signal. Differentially expressed NPRs and WRKYs may interact with other phytohormones, leading to the amelioration of watermelon Fusarium wilt. Conclusions: further understanding of gene expression patterns will pave the way for interventions that effectively control the disease.

## 1. Introduction

Watermelon (*Citrullus lanatus*) Fusarium wilt disease caused by *Fusarium oxysporum f. sp. niveum* (FON) poses a serious threat to watermelon quality and yield [1,2,3]. Symptoms such as rotted, discolored roots and brown vascular bundles appeared in Fusarium wilt diseased plants [4]. Different techniques include chemical control [5], biological control [6,7], grafting [8], and the use of disease-resistant cultivars [9,10] are utilized to overcome this disease. Many researchers found that different cultivation systems, such as intercropping [2,11], companion cropping [12], and rotated cropping [1], represent safe and efficient methods to control soil-borne diseases. Previously, researchers have focused on mutual recognition and recognition competition between the host plants, pathogens, pathogenic factors, and host plant defense factors. However, the molecular mechanisms involved therein remain unknown. Most researches have already identified the vital role of salicylic acid (SA) in plant defense [4,13] and programmed cell death [14]. For instance, reviews highlight the necessity of isochorismate synthase (ICS) [15] and phenylalanine ammonia-lyase (PAL) genes to synthesize SA and signal transduction mechanisms for activating the hypersensitive response and mediating system-acquired resistance (SAR) [13,16,17]. SA is also the main activator of defense response signal transduction [13,18]. Dou et al. determined that the expression of the genes involved in SA signal transduction increased, including the PR-1, WRKY, and TGA transcription factors, indicating that these genes play key roles in Foc4 infection [19]. Once certain levels of SA have been generated, SA-mediated activation of pathogen-related (PR) genes leads to a feedback in ICS expression in Arabidopsis, therefore preventing excessive SA accumulation [20,21]. Moreover, other studies show that the application with low level of exogenous SA could enhance plant resistance, while high concentration of it is harmful to plants [22,23]. Recently, Wang found that exogenous SA concentration >200 μM inhibited Foc TR4 growth [24]. Lv noticed that FON infection increased the accumulation of SA in watermelon [11]. Ren reported that exogenous SA (160 mg/L) could inhibit FON sporulation, but exogenously applied 100 μM of SA stimulate β-1,3-glucanase activity in watermelon leaves after FON inoculation [8]. Notably, our recent work revealed the important role of SA, JA, and ABA in resistance to watermelon Fusarium wilt. In addition, the results indicated that the SA content in the susceptible watermelon group was significantly higher than that in the resistant group [25].

Therefore, in this experiment, we aimed to exploring the role of SA in watermelon resistance to Fusarium wilt based on the expression difference of SA-related biosynthesis genes under different growth systems. Furthermore, our new findings provide a theoretical foundation for the SA regulation of watermelon Fusarium disease.

## 2. Results

### 2.1. Comparison of Disease Incidence, Watermelon Root Morphological Phenotype, and Physiological Indexes

In order to clarify the effect of exogenous SA on watermelon Fusarium disease resistance, the phenotypes of watermelon seedlings and the disease incidences after SA application at different concentrations at 7 dpi were explored. The observation of watermelon seedling results showed that there were severe plant yellowing and wilting symptoms in control group (0 μM, without SA treatment), while there were more leaves and developed roots after exogenous SA application at 100 μM as compared with it. However, a higher concentration of SA (200 μM) suppressed plant growth, shown by as the lower amount of leaves and roots of plants compared with 100 μM group (Figure 1A). In addition, the disease incidence in control group (0 μM, without SA treatment) was about 45%, but only 5% after exogenous SA application at 100 μM, which indicated the exogenous SA application could control the watermelon Fusarium disease occurrence (Figure 1B). Moreover, we compared the SA content after FON treatment with the control group in watermelon roots under the same growth conditions in the laboratory. The results showed that there was a significant increase in SA content in plants at 3 dpi and a descent after stabilization, but with a higher enrichment at the onset stage (7 dpi) (Figure 1C). Our data indicated that the SA was induced after FON attack in watermelon roots, suggesting that it may activate plant resistance at early stage. However, the reason for the SA accumulation at the onset stage might have been the disabled metabolisms or some modification activities [13].

Furthermore, the watermelon seedling morphology showed that the plants were much healthier and had more leaves and roots under rotation, as compared with those from the monocropping growth condition group (Figure 2A,B). Samples were referred to as C (continuous watermelon monocropping) and R (oilseed rape rotation cropping). The disease incidence results indicated that the incidence of watermelon Fusarium wilt in the rotated cropping system was significantly lower than that in the continuous cropping system group (Figure 2C). Moreover, the quantitative detection of FON in rhizosphere soil using real-time PCR [3,26] indicated that the abundance of FON exhibited no significant differences between these two systems (Figure 2D). This result suggested that there might be other reasons that led to plants being healthier in the rotated system, such as the structure of rhizosphere soil community [3,6,7]. The results from plant physiological indexes showed that the fresh root weight, root number, and root morphology were significantly affected. The fresh root weights (Figure 3A) in the R group were nearly twice those of the C group, and the pattern was the same for root number (Figure 3B). In addition, the results showed that there was a significant enhancement of the phenylalanine ammonia-lyase (PAL) enzyme activity (Figure 3D), but a decrease in peroxidase (POD) enzyme activity (Figure 3E) and malondialdehyde (MDA) content (Figure 3F) in the R group as compared with the C group. Therefore, these results showed that the plants under rotation (R) were healthier than those under the continuous cropping system (C). More importantly, our study showed a significantly higher SA content in the C group as compared to the R group, which confirmed that a higher level of SA was harmful to the plants (Figure 3C). Therefore, we aimed to further explore the different gene expression patterns related to SA pathways in watermelon roots that might be responsible for Fusarium wilt resistance under different cultivation systems.

### 2.2. Transcriptome Profiling of Watermelon Roots under Different Growth Conditions

We used transcriptome sequencing to analyze the molecular mechanisms of SA biosynthesis in watermelon plant roots activating Fusarium wilt resistance under different growth conditions. We randomly selected nine healthy plant roots using the five-spot sampling method from each greenhouse and mixed them as one replicate; we collected three replicates from three greenhouses. Samples were referred to as C (continuous watermelon monocropping) and R (oilseed rape rotation cropping). There were 21,910 genes detected and 5748 significantly deferentially expressed genes when comparing the C and R groups, with 3240 being upregulated and 2508 genes being downregulated (Figure 4A).

The Venn diagram and box plot show the list of expressed genes and their distribution under the two different systems (Figure 4B,C). The Venn diagram indicates that there were 14,245 genes were overlapped in C and R groups, and 392 significantly different expressed genes in C but 1671 significantly different expressed genes in R. The box plot shows that the gene expression distribution was uniform. The GO annotation analysis indicated that there were 17 significantly differentially expressed GO pathways compared under different groups in our experiment (Figure 4D). For instance, the metabolic process, localization, membrane, transporter activity, and catalytic activity (red markers in Figure 4D). The significantly expressed genes were merged into nine groups using COG classification, which mostly were in transcription (K) (Figure 4E). Further, the KEGG enrichment analysis revealed that five major pathways were significantly activated, including starch and sucrose metabolism, ribosome biogenesis in eukaryotes, endocytosis, cyanoamino acid metabolism, and especially phenylpropanoid biosynthesis (red marker in Figure 4F). Thus, our sequencing results supported the key role of SA in the resistance of watermelon to Fusarium wilt, indicating that it activated a series of defensive feedback mechanisms under environmental stress through several signal transduction pathways.

On the basis of the physiological indexes and gene function enrichment analysis, we analyzed the differential expression of various significantly expressed candidate genes related to pathogen resistance and SA biosynthesis. The results showed that there were increases in highly expressed genes in wound-induced proteins, peroxidases, and pathogenesis-related proteins in group C. However, genes related to stress-response proteins and other membrane-response proteins were highly expressed in group R (Figure 5A). Therefore, our results indicated that the continuous cropping environment may have induced the expression of certain peroxidases and pathogenesis-related genes, but the stress-response genes were not activated, which can lead to plant death. Moreover, our results illustrated that the main SA biosynthesis pathway in watermelon roots is the PAL pathway, and the upregulated expression of the CIPALs genes was an essential factor in SA accumulation in watermelon roots (Figure 5B).

### 2.3. Expression Verification of SA Biosynthesis Genes

In order to further test the hypotheses concerning the different expressed genes related to SA biosynthesis in watermelon roots under different growth conditions, we examined the results of the RT-qPCR analysis. These data confirmed significant differences in the expressions of the CIPALs, NPR, and NPR1 genes (Figure 6). The results emphasized the important role of the CIPAL1, CIPAL9, and CIPAL12 genes in SA biosynthesis. NPR expression was significantly higher in group R, while NPR1 expression was higher in group C, suggesting that they may have different functions under different growth conditions.

### 2.4. Bioinformatics Analysis of Candidate Genes

Three candidate SA synthesis genes were selected for protein homology analysis using NCBI BLAST. The results showed that the Cla008727 protein exhibited a high level of similarity with ABM 67591.1 | phenylalanine ammonia-lyase (*Vitis vinifera* L.) and XP_ 012082374.1 | phenylalanine ammonia-lyase (*Jatropha curcas* L.). The Cla013761 protein demonstrated a high level of similarity with *Cucumis sativus* L., *Ricinus communis* L., *Populus trichocarpa* (Torr. & Gray), *Populus tomentosa Carrière*, and *Cucumis melo* L. phenylalanine ammonia-lyase. In addition, the Cla 018303 protein had more than 90% similarity to many phenylalanine ammonia-lyase-like proteins in *Cucumis sativus* L. Therefore, our phylogenetic analysis showed that these candidate genes had the closest evolutionary relationship with *Cucumis sativus* L. and *Cucumis melo* L. (Figure 7). Six candidate genes related to SA metabolism were selected to analyze whether they possessed phytohormone cis-regulatory elements. Our results indicated that NPR and WRKY29 possessed an SA responsive element, whereas NPR1, WRKY, WRKY6, WRKY29, and WRKY72 possessed MeJA- and ABA-responsive elements (Table 1).

## 3. Discussion

Phytohormone SA is recognized as an effective defense signal against biotrophic pathogens [27,28]. Recent studies suggested that SA is related to at least three major strategies to disrupt SA-mediated defense [22,24,29], and that the cell survival or death decision is modulated through the temporal expression of SA at the infection site [30,31]. Researchers have carried out a large number of studies to elucidate the molecular role of SA in the immune response and dynamic regulation of Arabidopsis [21,32]. Similarly, most studies demonstrated that SA synthesis is induced under stress conditions in watermelon. For instance, Cheng et al. found that low temperatures induced salicylic acid production in watermelon via the phenylalanine pathway, which may coordinate with the redox signal to regulate resistance [33]. Lv et al. reported that a wheat–watermelon intercropping system inoculated with FON induced salicylic acid synthesis in watermelon [11]. Guang et al. noted that exogenous jasmonic acid (JA), SA, and ethylene (ET) significantly increased the expression of the CIOPR gene in watermelon [34]. However, the mechanisms involved in SA-induced watermelon immune responses to Fusarium wilt remain unclear. Therefore, in this study, we focused on the role of SA in watermelon resistance to FON under two treatments: rotating and nonrotating cropping. The reason why we used only one variety in this study is that the Zaojia 8424 variety of watermelon is the main cultivated variety in the Chinese market, and we have performed much field experiment research in these years [3,6,7].

Researchers have reported that the peroxidation of membrane lipids was activated when plants face environmental stresses [13,35]. For instance, POD is closely related to plant disease resistance. PAL is considered to be the plant defense enzyme [36]. It is closely related to the synthesis of various secondary metabolites, such as lignin, isoflavone phytoalexin, and flavonoid pigment, which contribute to disease resistance [4]. Moreover, MDA is a product of membrane lipid peroxidation, and its concentration indicates the degree of lipid peroxidation and membrane system injury [37]. Therefore, we measured the POD and PAL enzyme activities and the amount of MDA in watermelon plants under different growth conditions to determine the degree of stress. The biochemical analysis revealed that the significantly reduced POD activity and MDA content, as well as the increased PAL activity, may account for the healthier plants under rotated cropping. Furthermore, the RNA-seq and RT-qPCR results can both explain the significantly lower PAL enzyme activity, as demonstrated by the upregulated expression of the CIPAL1 and CIPAL2 genes in the C group as compared to the R group. Moreover, certain genes, such as those coding for wound-induced proteins (Cla006613, Cla007723, Cla007724) and peroxidase (Cla015181, Cla015182, Cla016910, Cla016911, Cla016937, Cla017829, Cla003189), demonstrated a lower expression in the R group, which may have resulted in lower enzyme activities related to POD and MDA contents.

Firstly, we confirmed that SA played a key role in improving watermelon resistance to Fusarium wilt in the pot experiment. Thereafter, we further demonstrated the significant accumulation of SA in watermelon plants under both types of cropping systems, which suggested SA has a function in responding to environmental stress. Thus, these results indicated that lower concentrations of exogenous SA effectively decreased watermelon Fusarium wilt incidence, suggesting that SA may help activate disease resistance [13,22]. Wildermuth et al. reported that loss of ICS1 abolishes pathogen-induced SA accumulation [15]. Arabidopsis with mutations in the PAL genes exhibited a 90% reduction in basal PAL activity and a 50% decrease in pathogen-induced SA accumulation [13,27]. On the contrary, the IC pathway seemed to be the major route for SA biosynthesis and immunity [17,38]. Silenced PAL genes were also shown to cause a loss of SA induction in soybean upon pathogen infection [27]. Differential PAL activity related to SA induction was reported in tobacco [23], tomato [18], and cucumber [17], raising questions as to whether different plants use these two pathways for pathogen-induced SA synthesis in different ways. Lefevere wrote that the importance of both pathways for biosynthesis differed between plant species, rendering it difficult to make generalizations about SA production that cover the entire plant kingdom [39]. After analyzing the differentially expressed genes, the results showed that there was significant upregulated gene expression of CIPAL1, CIPAL2, and CIPAL12 related to SA synthesis in the rotated cropping systems, with only CIPAL9 being significantly more expressed in the continuous cropping treatment. The higher relative expression of the CIPALs genes in plants in the continuous cropping system might explain why there was more SA accumulation.

After SA is synthesized, plant pathogens deploy diverse mechanisms to interfere with SA downstream signaling. NPR1 is considered to form a complex with the SA receptor [16,40]. Therefore, in order to identify the regulatory components downstream of SA in watermelon, we analyzed the NPR family gene expression under different growing conditions. However, the results from the differentially expressed NPR family genes in the two cropping systems indicated that they may play different roles in active downstream signaling of watermelon disease resistance responses. SAR is an induced broad-spectrum immune mechanism in plants [21]. Thus, our results indicated that the gene expression related to stress-responsive proteins and other membrane-responsive proteins at low levels of SA in watermelon roots suggested that there might be an associated between NPR with SAR in ensuring watermelon plant health. Contrarily, it was reported that activated immune responses, such as the production of reactive oxygen species, the activation of MAP kinases, and transcriptional reprogramming, occur in both pathogen-associated molecular pattern-triggered immunity (PTI) and effector-triggered immunity (ETI) [41]. In addition, ETI often leads to programmed cell death at the site of infection [42]. Therefore, our results indicated that pathogen-related responses were associated with ETI, and certain wound-induced responses led to plant cell death when highly accumulated via NPR1 SA in watermelon roots. Collectively, the differentially expressed genes induced SA biosynthesis and activated the Fusarium wilt defense response in watermelon, which explained the different disease incidences under the two growth conditions.

Moreover, increasing evidence suggests that WRKY transcription factors play an essential role in plant defense to pathogen infection [43,44]. In Arabidopsis thaliana, WRKY70 was reported to be a mediator of jasmonic acid (JA) suppression and abscisic acid (ABA) responses by SA [45]. Rice NPR1 and WRKY13 were also mediated by SA to suppress the JA response [20]. In addition, our RNA-seq and RT-qPCR results identified some significantly differently expressed WRKY transcription factors (Figure 5B and Figure 6) under different growth conditions. Furthermore, the candidate gene promoter prediction showed that NPR1, WRKY, WRKY6, WRKY29, and WRKY72 contained cis-acting elements related to other plant hormones (MeJA and ABA). Therefore, we suggested that SA may play an important role in triggering the watermelon plant immune system and crosstalk with other phytohormones.

The key question concerned whether the rotation changed the soil microbiome and fertilization, which ultimately reduced the soil contamination, or whether the phytohormone networks mediating the watermelon plant immune system increased resistance to FON, resulting in a lower disease incidence. Recent studies demonstrated that the plant immune system shapes the microbiome, which can increase the plant immune capacity [46,47,48]. Therefore, we are now working toward revealing the mechanism of interaction between SA and rhizosphere microorganisms in watermelon resistance to Fusarium wilt.

## 4. Materials and Methods

### 4.1. Experimental Site Description and Sampling

The pot experiment was conducted in the city of Changsha (112°58′42″ E, 28°11′49″ N), Hunan Province, China. The soil was sandy loam with background sterilization (LDZM-80KCS-3 vertical pressure steam sterilizer, ZHONGAN, Shanghai, China) before use [3]. The trial watermelon variety was Zaojia 8424 (Xinjiang Farmer Seed Technology Co., Ltd., Urumqi, China), which is the main cultivated variety on the Chinese market. The watermelon seedling nutrition bowl was cultivated and grown in a biochemical incubator (LRH-300, ZHUJIANG, Taihong, Shaoguan, China) at Tm 25 °C, light 16 h/Tm 18 °C, dark 8 h. The nutrition bowl seedling substrate included peat, perlite, and vermiculite (6:3:1). We transplanted each plant into pots separately after 30 days. One experiment concerned a study of the effect of exogenous SA on watermelon Fusarium disease resistance, and screening of the optimal concentration of SA to improve watermelon seedling growth. The 5 mL exogenous SA (Sigma-Aldrich LLC., Merck KGaA, Darmstadt, Germany), with a gradient of 0 μM, 10 μM, 50 μM, 100 μM, and 200 μM, was incorporated into the root zone of each plant, and 5 mL was added again 24 h later. Two days after the watermelon seedlings reached the stage of possessing two leaves per plant, 5 mL aliquots of 10^6^ conidia/mL FON were added into the root zone of each plant [2]. Thereafter, the phenotypes of the watermelon seedling were compared, and disease incidences at 7 dpi (7 days postinoculation) were recorded. Moreover, in order to identify the effect of SA content after FON incubation, 5 mL aliquots of 10^6^ conidia/mL FON were incorporated into the root zone of each watermelon plant when they possessed two leaves per plant, and watermelon root samples were collected at different sampling times. Samples were referred to as S: control (Zaojia 8424, sensitive cultivar); and SF (Zaojia 8424, sensitive cultivar +FON). Moreover, we used 0 dpi (before treatment); 12 hpi (12 h postinoculation); 1 dpi (1 day post inoculation); 3 dpi (3 days postinoculation); 5 dpi (5 days postinoculation); and 7 dpi (7 days postinoculation). We selected 15 watermelon plants as one repetition, and set three independent replicates for each sample group.

The field experiment was conducted at the Gaoqiao Scientific Research Base of the Hunan Academy of Agricultural Sciences in Changsha (112°58′42″ E, 28°11′49″ N) [3]. The soil was sandy loam. The greenhouse experiment was designed to mimic actual plant growth conditions. It was carried out in six greenhouses (30 m × 6 m) with the same temperature and light conditions. Plants were left in the greenhouses for over a year after harvesting, then we designed two cultivation modes: continuous watermelon monocropping and rotated with oilseed rape cropping. The soil for monocropping watermelon (the diseased soil) had been cultivated for 5 years, and was used as a positive control in the experiment. The disease incidences of Fusarium wilt were all above 80% since watermelon monocropping in year 2016 (Appendix A). The trial watermelon variety was Zaojia 8424, which was provided by Xinjiang Farmer Seed Technology Co., Ltd., China. The routine cultivation management in the greenhouses was the same. The watermelon seedling cultivated nutrition bowls were transplanted into a greenhouse after 30 days. The vines were pruned to enable climbing, and a planting density of 360 plants per 667 m^2^ was maintained. We randomly selected nine healthy plants by using the five-spot sampling method from each greenhouse and mixed them as one replicate; we collected three replicates from three greenhouses. Samples were referred to as C (continuous watermelon monocropping) and R (oilseed rape rotation cropping). Samples were sealed into labeled bags and taken back to the laboratory immediately within an ice box. We divided them into four groups for determining the physiological indexes, biochemical indexes, and other molecular experimental analyses, and stored them at −80 °C.

### 4.2. Detection and Incubation of FON in Plant Samples

The ITS sequence of the 5.8S rDNA of the nine isolates was amplified and sequenced. The sequence was deposited in GenBank (BankIt2435044 BSeq#1 MW700270). We also used the EF-1α gene to identify the isolated Fusarium fungus, which was BLASTn to the Fusarium-ID database. The sequence was deposited in GenBank (Bank-It2479731 MZ540776). The BLAST results confirmed the identity of the isolates as *Fusarium oxysporum f. sp. niveum*. The Fusarium strain FON was firstly incubated in the dark for 7 days on PDA at 28 °C. Then, a bam plug was selected from a PDA plate and placed into 300 mL of potato dextrose broth in a flask, before propagation on a rotary shaker at 200 rpm at 26–30 °C.

### 4.3. Disease Investigation

The incidence of Fusarium wilt was defined as being when the obvious symptoms appeared, such as plant yellowing, wilting, and the FON identification test in 4.2. The disease incidence (%) = (no. of infected plants/total number of plants surveyed) × 100%.

### 4.4. Soil DNA Extraction and Quantitative Detection of FON by Real-Time PCR

For each replicate, nine independent rhizosphere soil samples were pooled from watermelon plant roots from each greenhouse. Three greenhouses within the same treatment group were regarded as three independent replicates. Samples were referred to as C (continuous watermelon monocropping) and R (oilseed rape rotation cropping). The Power Soil DNA isolation kit was used for the extraction of genomic DNA according to the manufacturer’s instructions (Omega Bio-Tek, Inc., Norcross, GA, USA). Distinct regions of the FON rRNA genes were amplified by PCR (Bio-Rad T100TM Thermal Cycler, Bio-Rad Laboratories, Inc., Hercules, CA, USA) using specific primers (Fonq-F (5′-GTTGCTTACGGTTCTAACTGTGC-3′), Fonp1-R (5′-CTGGTACGGAATGGCCGATCAG-3′)). A 1446 bp DNA fragment containing the qPCR target sequence was amplified from soil DNA using conventional PCR (initial incubation at 94 °C for 4 min, followed by 18 cycles of 94 °C for 40 s, 60 °C for 40 s, 72 °C for 70 s, and a final extension at 72 °C for 10 min [11,26]. 

### 4.5. Determination of Physiological and Biochemical Indexes of Watermelon Plant Roots

We randomly selected nine healthy plants from each greenhouse using the five-spot sampling method and mixed them as one replicate; we collected three replicates from three greenhouses. Samples were referred to as C (continuous watermelon monocropping) and R (oilseed rape rotation cropping). Firstly, we collected each watermelon root with sterilized scissors in order to measure the fresh root weights with an electric balance, and then we counted the number of roots. The peroxidase (POD) activity and phenylalanine ammonia-lyase (PAL) activity of the plant samples were analyzed using the A084-3-1 Peroxidase assay kit and the A137-1-1 PAL test kit, respectively (Nanjing Jiancheng Bioengineering Research Institute, Nanjing, China), according to manufacturer’s protocol. The malondialdehyde (MDA) content was determined by the thiobarbituric acid method with an A003-1-2 MDA assay kit (Nanjing Jiancheng Bioengineering Research Institute, Nanjing, China) according to manufacturer’s protocol. A Tecan-SPARK (Tecan Trading AG, Männedorf, Switzerland) and Eppendorf 5415R refrigerated centrifuge (Eppendorf AG, Hamburg, Germany) were used to test the enzyme activities. Three biological replicates per sample with three technical replicates were performed.

### 4.6. Determination of SA Content

The SA content was measured using liquid chromatography–tandem mass spectrometry as follows: A fresh sample (200 mg) was frozen with liquid nitrogen and homogenized using a Tissue Lyser homogenizer (Qiagen, Germantown, Germany). Thereafter, 1 mL of 80% methanol was added, and the homogenates were mixed in an ultrasonic bath and maintained overnight (4 °C). The supernatant was collected (using centrifugation at 15,000× *g* for 10 min) and vacuumed to dry in a Jouan RCT-60 concentrator (Jouan, Saint-Herblain, France). Then, the dried extract was dissolved in 200 μL of sodium phosphate solution (0.1 mol L-1, pH 7.8) and passed through a Sep-Pak C18 cartridge (Waters, Milford, MA, USA) eluted with 1.5 mL of 80% methanol. After vacuuming to dry, and the eluate was dissolved in 10 mL of 10% methanol, and 5 μL of the solution was injected into the liquid chromatography–tandem mass spectrometry system (LCMS-8030, Shimadzu Corporation, Chiyoda-ku, Japan) [49]. Three biological replicates per sample with three technical replicates were performed. The determination system for the SA content in watermelon roots is shown in Appendix A. 

### 4.7. RNA Extraction, Library Preparation, and Illumina Novaseq 6000 Sequencing

To examine the molecular mechanisms of Fusarium wilt resistance in watermelon plants, we used transcriptome sequences to analyze the watermelon roots under different growth conditions. The samples were referred to as C (continuous watermelon monocropping) and R (oilseed rape rotation cropping). We randomly selected nine healthy plant roots from each greenhouse using the five-spot sampling method and mixed them as one replicate; we collected three replicates from three greenhouses. At total of six RNA samples were extracted from different groups of watermelon plant roots using the TRIzol Plant RNA Purification Reagent (Invitrogen, Inc., Carlsbad, CA, USA) according to the manufacturer’s instructions; genomic DNA was removed using DNaseI (Takara Bio Inc., Caojin City, Sihe County, Japan). The RNA concentration and purity were determined using a NanoDrop 2000 UV–vis spectrophotometer (Thermo Scientific, Wilmington, DE, USA), and checked using 1% agarose gel electrophoresis (EPS-300, TANON Science & Technology Co., Shanghai, China). Thereafter, RNA purification, reverse transcription, and library construction and sequencing were performed according to the manufacturer’s instructions (Shanghai Majorbio Bio-Pharm Technology Co., Ltd., Shanghai, China). The insert size of the library was detected using an Agilent 2100 bioanalyzer with an RNA concentration > 200 ng/uL, RNA Integrity Number (RIN) ≥ 8.0, OD260/280 ≥ 1.8, and OD260/230 ≥ 1.5. The RNA-seq transcriptome library was prepared following the TruSeq RNA sample preparation kit (Illumina, San Diego, CA, USA). Libraries were size-selected for cDNA target fragments of 200–300 bp on 2% low-range ultra-agarose, followed by PCR amplification using Phusion DNA polymerase (NEB) for 15 PCR cycles. After quantification using TBS380, the paired-end RNA-seq sequencing library was sequenced with the Illumina Novaseq 6000 (2 × 150 bp read length). Three biological replicates per sample were analyzed. The clean reads were deposited into the NCBI Sequence Read Archive database (Accession Number: PRJNA641525). The reference gene version is watermelon 97103 in the Cucurbit Genomics Database (http://cucurbitgenomics.org/organism/1, accessed on 12 November 2021). We selected Hisat2 v2.0.5. as the mapping tool based on the gene model annotation file.

To identify differential expression genes (DEGs) between the two groups of samples, according to the calculated fragments per kilobase of exon per million mapped reads (FPKM), EdgeR [50] software was utilized. Furthermore, functional-enrichment analysis, including GO (gene ontology) functional enrichment, COG (clusters of orthologous groups), and KEGG (Kyoto Encyclopedia of Genes and Genomes) pathway analyses were performed.

### 4.8. Quantitative Detection of Candidate Genes by RT-qPCR

The plant roots’ RNA was extracted using TRIzol Plant RNA Purification Reagent (Invitrogen, Inc., Carlsbad, CA, USA). A Promega GoScript Reverse Transcription System (Promega Biotech Co., Madison, WI, USA) was used for obtaining cDNA. The final cDNA quality was checked using a NanoDrop 2000 UV–vis spectrophotometer (Thermo Scientific, Wilmington, DE, USA), and 2% agarose gel electrophoresis (EPS-300, TANON Science & Technology Co., Shanghai, China) with a concentration > 150 ng/μL. Distinct regions of candidate rRNA genes were amplified by PCR (Eppendorf AG, Bio-Rad Laboratories, Inc., Hercules, CA, USA) using specific primers (Appendix A). Then, the PCR products were used as templates to construct the standard curve of the fluorescence quantitative PCR (Bio-Rad CFX96 Real-Time System, Bio-Rad Laboratories, Inc., Hercules, CA, USA) using each primer. The correlation coefficient (R^2^) of CIACT was 0.989, demonstrating a PCR efficiency of 103.9%. The correlation coefficients (R^2^) of the other candidate genes were more than 0.98, and the PCR efficiency was between 90–110%. The expression levels were calculated using the 2-ΔCT method. Primer Premier 5.0 (Premier, Inc., Vancouver, BC, Canada) was used to design family-specific primers. For the sequences with a high homology, dnaman 7.0 (Lynnon Biosoft, San Ramon, CA, USA) was used for multiple sequence alignment, and primers were designed in the nonconservative region. CIACT was used as a reference gene [11].

### 4.9. Statistical Analysis

Statistical analysis was performed using GraphPad Prism 9 (GraphPad Software, San Diego, CA, USA). Tukey’s test, followed by Student’s *t*-test (*p* ≤ 0.05), was used to analyze the significant differences between the two groups. All values are expressed as mean ± standard error (*n* = 3). The figures were constructed by using Microsoft Office 2010 (Microsoft Corporation, Redmond, WA, USA).

## 5. Conclusions

Our experiment confirmed the essential role of SA in watermelon plant defense activation against Fusarium wilt disease under different growth conditions. Our results indicated that the expression of the CIPALs genes was key to SA accumulation in watermelon plants. The NPR family genes may have played different roles in responding to the SA signal. Moreover, these differentially expressed WRKY transcription factors interacted with other phytohormones, such as ABA and JA, which led to the amelioration of watermelon Fusarium wilt. Our new findings provide a theoretical foundation for the SA regulation of watermelon Fusarium disease.

## Figures and Tables

**Figure 1 plants-11-00293-f001:**
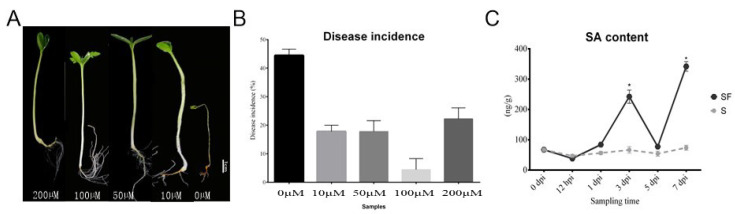
The effect of salicylic acid on watermelon resistance to Fusarium wilt in the pot experiment: (**A**) The phenotypes of watermelon seedling after SA application at different concentrations; (**B**) comparison of disease incidence after different concentrations of exogenous SA were applied; (**C**) comparison of SA content in watermelon treated with FON. S: control (Zaojia 8424, sensitive cultivar); SF: Zaojia 8424, sensitive cultivar + FON. Note: 0 dpi (before treatment); 12 hpi (12 h postinoculation); 1 dpi (1 day postinoculation); 3 dpi (3 days postinoculation); 5 dpi (5 days postinoculation); 7 dpi (7 days postinoculation). Data are expressed as mean ± SE (*n* = 3). Multiple t tests of two-way ANOVA (* *p* ≤ 0.0001).

**Figure 2 plants-11-00293-f002:**
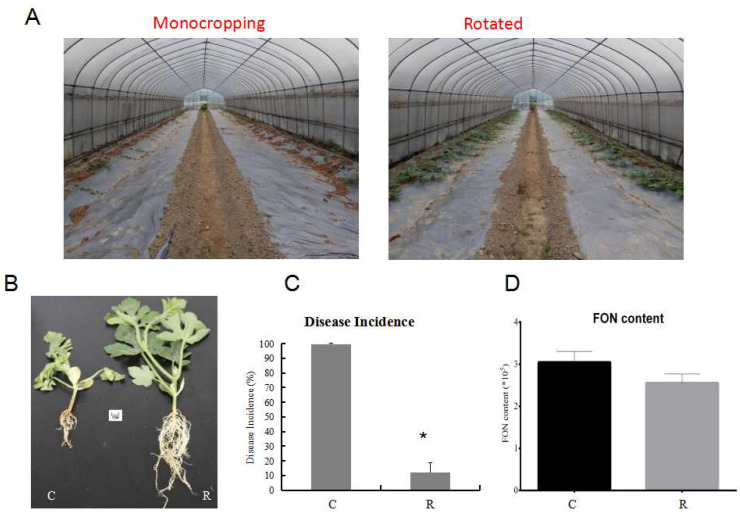
Comparison of watermelon growth and disease incidence under different growth conditions in the field experiment. (**A**) Comparison of watermelon plants’ morphology under monocropping and rotated cropping systems; (**B**) the phenotypes of watermelon seedlings under different systems; (**C**) the disease incidence under different growth systems; (**D**) FON biomass in two cropping systems. C: continuous watermelon monocropping; R: oilseed rape rotation cropping. Data are expressed as mean ± SE (*n* = 3). Student’s *t*-test (* *p* ≤ 0.05).

**Figure 3 plants-11-00293-f003:**
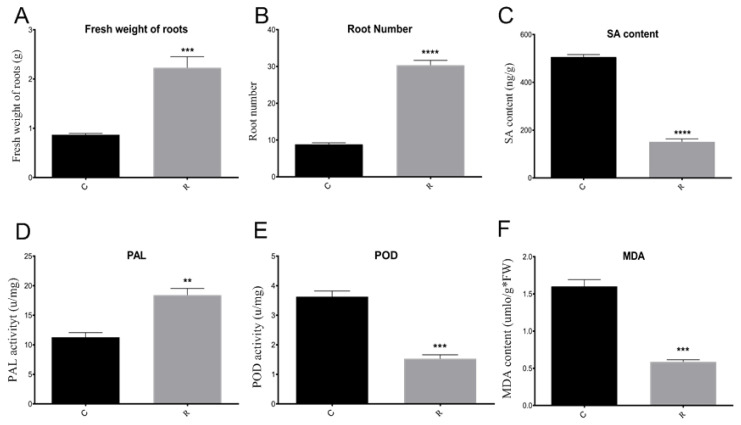
Comparison of watermelon physiological and biochemical indexes. (**A**) Comparison of fresh root weights in different samples; (**B**) comparison of root number in different samples; (**C**) comparison of SA contents in different samples; (**D**) comparison of PAL enzyme activity in different samples; (**E**) comparison of POD enzyme activity in different samples; (**F**) comparison of MDA content in different samples. C: continuous watermelon monocropping; R: rotated with oilseed rape cropping; POD: peroxidase; PAL: phenylalanine ammonia-lyase; MDA: malondialdehyde. Three biological replicates per samples were analyzed. Data are expressed as mean ± SE (*n* = 3). Student’s *t*-test (** *p* ≤ 0.01; *** *p* ≤ 0.001; **** *p* ≤ 0.0001).

**Figure 4 plants-11-00293-f004:**
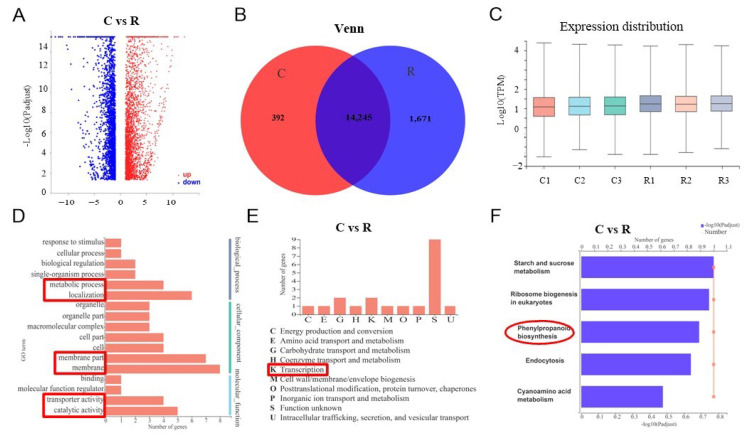
Transcriptome comparison analysis of watermelon gene expression under different cropping systems: (**A**) volcano map with different expressions; (**B**) Venn diagram of watermelon genes expression under different cropping systems; (**C**) box plot of gene expression distribution; (**D**) GO annotations analysis; (**E**) COG classification; (**F**) comparison analysis of major KEGG pathway enrichment. C: continuous watermelon monocropping; R: oilseed rape rotation cropping. Three biological replicates per samples were analyzed.

**Figure 5 plants-11-00293-f005:**
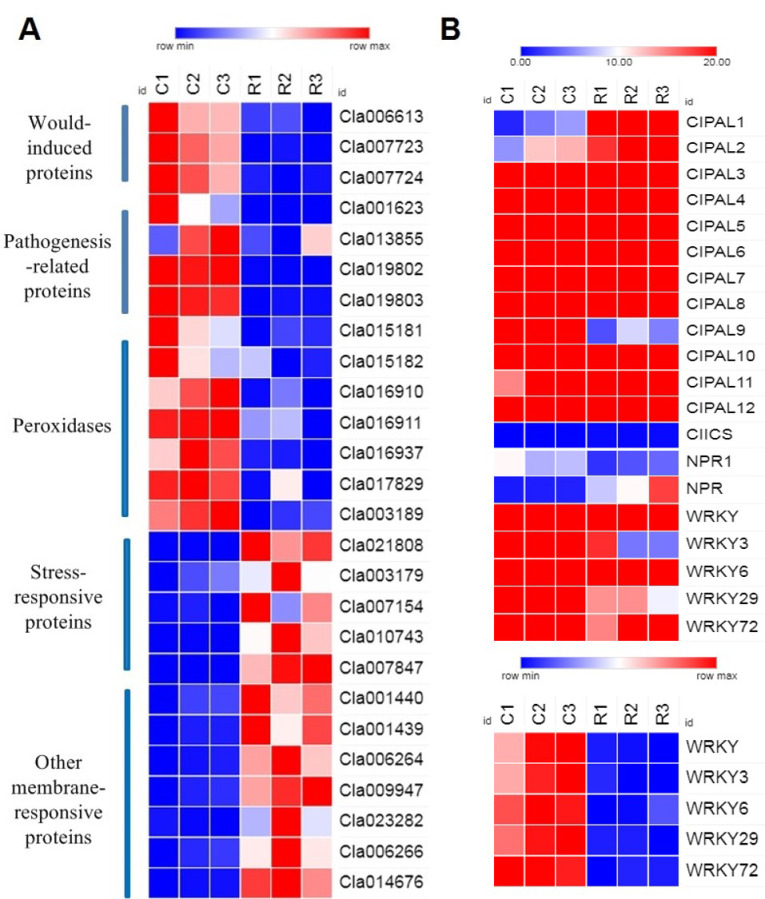
Differential expression genes in watermelon roots under different cropping systems: (**A**) comparison analysis of the expression of 26 genes under different cropping systems; (**B**) comparison analysis of the relative expression of 20 candidate genes in SA biosynthesis. The relative color scheme uses selected values in each row to convert values to colors. C: continuous watermelon monocropping; R: oilseed rape rotation cropping. Three biological replicates per samples were analyzed.

**Figure 6 plants-11-00293-f006:**
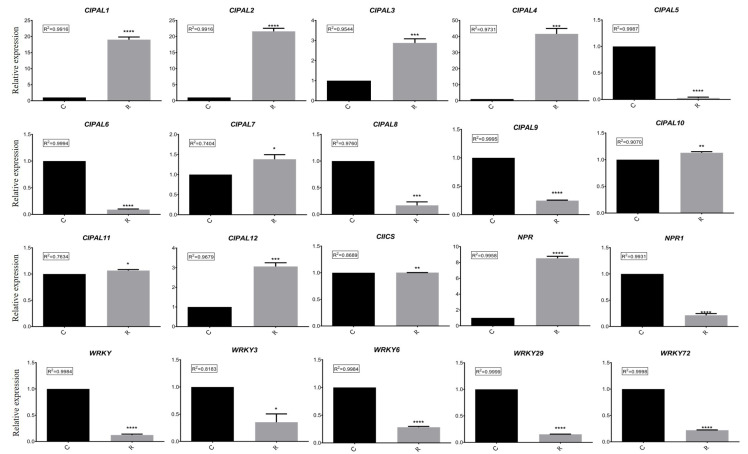
Comparison analysis of relative expressions of 20 candidate genes in different samples using RT-qPCR. C: continuous watermelon monocropping; R: oilseed rape rotation cropping. Three biological replicates per samples were analyzed. Data are expressed as mean ± SE (*n* = 3). Student’s *t*-test (* *p* ≤ 0.05; ** *p* ≤ 0.01; *** *p* ≤ 0.001; **** *p* ≤ 0.0001).

**Figure 7 plants-11-00293-f007:**
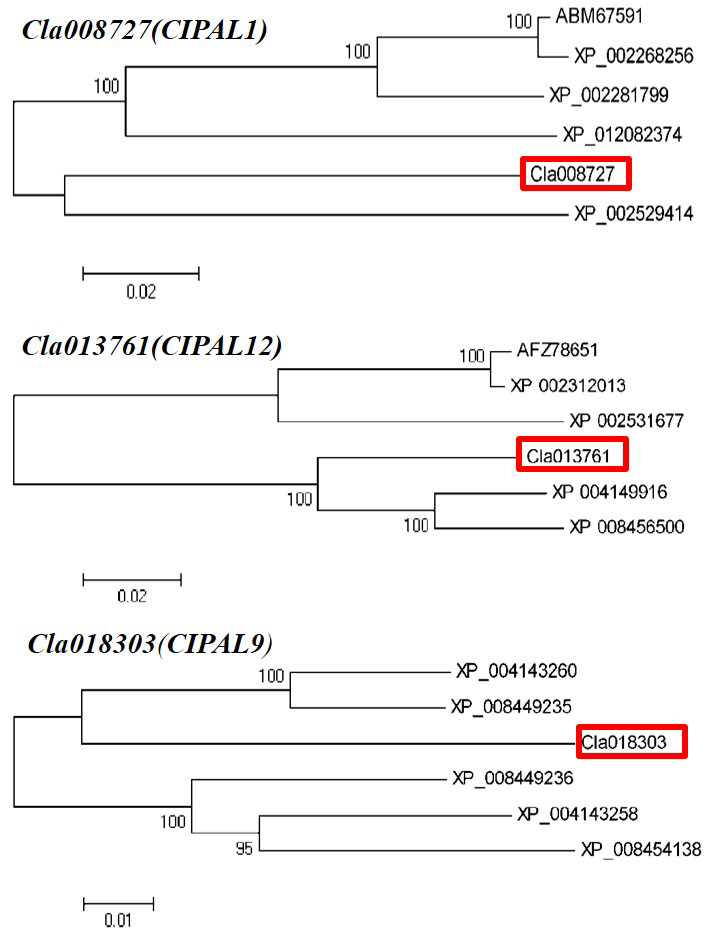
Phylogenic tree of the homologue’s proteins from three candidate genes.

**Table 1 plants-11-00293-t001:** Prediction of cis-acting regulatory element of six DEGs.

Gene ID	Gene Name	Cis-Acting Element	Function	Sequence	Number
Cla002899	*NPR1*	ABRE	ABA	ACGTG	1
Cla019154	*NPR*	CGTCA-motif	MeJA	CGTCA	2
		TCA-element	SA	CCATCTTTTT, TCAGAAGAGG	5
Cla022362	*WRKY*	ABRE	ABA	AACCCGG	1
		CGTCA-motif	MeJA	CGTCA	1
Cla002084	*WRKY6*	CGTCA-motif	MeJA	CGTCA	1
Cla005515	*WRKY2*	ABRE	ABA	CACGTG, ACGTG	5
		CGTCA-motif	MeJA	CGTCA	2
		TCA-element	SA	CCATCTTTTT	1
Cla010867	*WRKY72*	ABRE	ABA	ACGTG	2
		CGTCA-motif	MeJA	CGTCA	3

## Data Availability

The clean reads were deposited into the NCBI Sequence Read Archive (SRA) database (Accession Number: PRJNA641525). The BLAST result confirmed the identity of the isolates as *Fusarium oxysporum f. sp. Niveum*. The sequences were deposited under the GenBank: BankIt2435044 BSeq#1 MW700270, BankIt2479731 MZ540776.

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
