# Peer review of "Study on the Role of Salicylic Acid in Watermelon-Resistant Fusarium Wilt under Different Growth Conditions"

_plants, 2022, doi:10.3390/plants11030293_

Round 1

Reviewer 1 Report

In this manuscript, the authors showed that there were more leaves and developed roots, and lower disease incidence after exogenous SA application at 100 μM. The disease incidence was also lower in rotated cropping systems than in monocropping, but with a lower SA content.

  1. This manuscript analysis the transcriptome profiling of watermelon roots under C (continuous watermelon monocropping) and R (oilseed rape rotation cropping). In order to understand whether SA mediating the watermelon plant immune system increase resistance to FON resulting in a lower disease incidence, the transcriptome profiling of watermelon roots under SA treatment (100 μM) and its control group (0 μM, without SA treatment) showed also be analyzed.
  2. The BA2H gene in this manuscript was selected the Cla015062 protein, according to the high level of similarity with probable E3 ubiquitin-protein ligase BAH1-like 1 protein sequences. Do any references have indicated these two proteins had the same gene function? If yes, please add a related reference. As far as I know, no BA2H gene has been identified in the plants yet.
  3. In Figure 1C, please check again whether there is really no significant difference between SF and S at 3 dpi and 7 dpi.
  4. Please add the LCMS chromatograms of standard compound (SA) and sample extracts

Author Response

Dear reviewer,

Thanks for all your kindly suggestion. All the suggested revisions involve minor changes that have been incorporated into the revision and marked up using the ”Track Changes” function. The point by point, the details of the revisions to the manuscript and our responses to the referees’ comments are in the attachment file.

Point 1: This manuscript analysis the transcriptome profiling of watermelon roots under C (continuous watermelon monocropping) and R (oilseed rape rotation cropping). In order to understand whether SA mediating the watermelon plant immune system increase resistance to FON resulting in a lower disease incidence, the transcriptome profiling of watermelon roots under SA treatment (100 μM) and its control group (0 μM, without SA treatment) showed also be analyzed.

Response 1: Thanks for your kindly suggestion. As you mentioned, for proving biosynthesis of SA is effective in defending Fusarium wilt, we have also compared the differences of watermelon seedling and root phenotype, defense enzyme activities, SA content, and the SA signal pathway related genes expression from different aspects to confirm the effect of SA on improving watermelon resistance to Fusarium wilt. From the diversities among resistance/ susceptible cultivars, exogenous SA application, pathogen infection in the pot experiment. The physiological and biochemical indexes results indicated that the SA content and PAL enzyme activity were significantly increased in watermelon root after FON incubation compared with non-inoculated group under the same growth condition. Besides, we have compared the physiological and biochemical indexes of resistant and susceptible watermelon varieties after FON incubation, established the SA, JA, and ABA determination system in watermelon roots, and used transcriptome sequencing and RT-qPCR to identify the differential expressed genes in different varieties related to phytohormones signaling pathways at onset stage to study the role of SA, JA, and ABA in resistance to watermelon Fusarium Wilt. The results indicated that the main pathway of SA biosynthesis in watermelon roots is PAL pathway, and the CIPALs and BA2H genes are essential factors lead to SA accumulation in watermelon. The opposite expression of SA receptor NPR genes may play different roles in watermelon disease-resistance response. And this part of research has been accepted (plants 1514768) and will be online soon.

Since recent studies have shown that some pathogenic effectors have evolved in SA biosynthesis pathway and signal transduction, which playing key roles to mediating plant immune system. And our previous work identified the relationship between rhizosphere community structure and the occurrence of watermelon Fusarium wilt. We therefore are working on exploring the mechanism of how salicylic acid signal recruit soil microorganism to against Fusarium wilt disease by metagenomic. This research has identified the application of SA changing the watermelon rhizosphere soil microbial communitie, and provides new sight for microbial function in management and agricultural practices of plant healthy. And in this part of research, we have analyzed the watermelon roots under SA treatment (100 μM) and its control group (0 μM, without SA treatment) by transcriptome profiling to further understand whether SA mediating the watermelon plant immune system increase resistance to FON resulting in a lower disease incidence at different time point. Furthermore, we are now working out to further verify the DEG function based on CRISPER/CAS9-mediated gene editing technology in watermelon plants and published these data together as soon as possible.

Point 2: The BA2H gene in this manuscript was selected the Cla015062 protein, according to the high level of similarity with probable E3 ubiquitin-protein ligase BAH1-like 1 protein sequences. Do any references have indicated these two proteins had the same gene function? If yes, please add a related reference. As far as I know, no BA2H gene has been identified in the plants yet.

Response 2: Thanks for your kindly suggestion. We have rewritten the sentence and add the reference about the information of similarity with probable E3 ubiquitin-protein ligase BAH1-like 1 protein sequences in line 209-210.

Point 3: In Figure 1C, please check again whether there is really no significant difference between SF and S at 3 dpi and 7 dpi.

Response 3: Thanks for your kindly suggestion. We have updated the new Figure 1 with revised Figure 1C in line 83.

Point 4: Please add the LCMS chromatograms of standard compound (SA) and sample extracts.

Response 4: Thanks for your kindly suggestion. We have added the LCMS chromatograms of standard compound (SA) and sample extracts in supplementary material 2 figure S1.

Reviewer 2 Report

Please correct

Results

  1. 5, line 144: Figure 4 is not fully explicit (is to small, especially A, B and C). Please improve the figure.

Material and method

  1. 10, lines 323-324: Please eliminate the spaces between value and ° ….(112 °58’42’’E, 28 324 °11’49’’N)…
  2. 11, lines 335-337: Please rephrase and introduce the missing info

…gradient of exogenous SA (Sigma-Aldrich LLC., Merck KGaA, 335 Darmstadt, Germany), from 0 μM, 10 μM, 50 μM, 100 μM, to 200 μM, was incorporated into the root zone of each plant and 5 mL was added again 24 h later.

Which SA quantity was added at the beginning??? Info is missing

  1. 11, line 339: Please correct ….106 is 106 5 mL aliquots of 106 conidia/mL FON were….
  2. 11, line 352: Please eliminate the spaces between value and ° …(112°58’42’’E, 28 °11’49’’N)
  3. 11, line 353: Please delete ( from (The soil was sandy loam.

Author Response

Dear reviewer,

Thanks for all your kindly suggestions. 

All the suggested revisions involve minor changes that have been incorporated into the revision and marked up using the ”Track Changes” function. The point by point, the details of the revisions to the manuscript and our responses to the referees’ comments are in the attachment file.

Point 1: Results

1.5, line 144: Figure 4 is not fully explicit (is to small, especially A, B and C). Please improve the figure.

Response 1: Thanks for your kindly suggestion. We have revised the Figure 4 (especially A, B and C) in line 146.

Point 2: 10, lines 323-324: Please eliminate the spaces between value and ° .(112 °58’42’’E, 28 324 °11’49’’N).

Response 2: Thanks for your kindly suggestion. We have eliminated the spaces between value and ° .(112 °58’42’’E, 28 324 °11’49’’N). in line 324-325.

Point 3: 11, lines 335-337: Please rephrase and introduce the missing info. …gradient of exogenous SA (Sigma-Aldrich LLC., Merck KGaA, 335 Darmstadt, Germany), from 0 μM, 10 μM, 50 μM, 100 μM, to 200 μM, was incorporated into the root zone of each plant and 5 mL was added again 24 h later. Which SA quantity was added at the beginning??? Info is missing

Response 3: Thanks for your kindly suggestion. We have rewritten the sentence in line 335-336.

Point 4: line 339: Please correct ….106 is 106… 5 mL aliquots of 106 conidia/mL FON were…

Response 4: Thanks for your kindly suggestion. We have corrected the 106 as 106 in line 339.

Point 5: line 352: Please eliminate the spaces between value and ° …(112°58’42’’E, 28 °11’49’’N)

Response 5: Thanks for your kindly suggestion. We have eliminated the spaces between value and ° .(112 °58’42’’E, 28 324 °11’49’’N). in line 352.

Point 6: line 353: Please delete ( from (The soil was sandy loam.

Response 6: Thanks for your kindly suggestion. We have deleted the  “( from (The soil was sandy loam” in line 353.

Round 2

Reviewer 1 Report

The reference of "A chromosome-scale genome assembly of cucumber (Cucumis sativus L.) " could not provide that the BA2H gene is homology with the BAH1-like 1 gene. It was suggested to remove the content of the BA2H gene in this manuscript including in the abstract.
Also please give the m/z value of SA detected in LCMS in figure S1.

Author Response

Dear reviewer,

Thanks for all your kindly suggestion. All the suggested revisions involve minor changes that have been incorporated into the revision and marked up using the ”Track Changes” function.

Point 1: The reference of "A chromosome-scale genome assembly of cucumber (Cucumis sativus L.) " could not provide that the BA2H gene is homology with the BAH1-like 1 gene. It was suggested to remove the content of the BA2H gene in this manuscript including in the abstract.

Response 1: Thanks for your kindly suggestion. We have deleted the sentence with reference about the BA2H gene is homology with the BAH1-like 1 gene in line 210 and replaced the revised figure 7 without Cla 015062 protein. However, since many studies have identified the essential role of (benzoic acid-2-hydroxylase) BA2H genes in SA biosynthesis, we insist on retaining other contents of the BA2H gene in this manuscript.

Point 2: Also please give the m/z value of SA detected in LCMS in figure S1.

Response 2: Thanks for your kindly suggestion. We have added the m/z value of SA detected in LCMS in figure S1.

Round 3

Reviewer 1 Report

BA2H gene is so important in SA biosynthesis, but nobody has identified this gene sequence yet. The gene sequence in your manuscript might be wrong, which will be misleading readers. If you want to put the sequence of the BA2H in your manuscript, please do the enzyme activity in this manuscript.

Author Response

Dear reviewer,

Thanks for your kindly suggestion. Generally, we have removed the content of the BA2H gene in this manuscript including in the abstract and provided the revised figure 5, figure 6, and Table S2. Besides, we have referenced our new published related paper in the introduction part. All the suggested revisions involve minor changes that have been incorporated into the revision and highlighted in yellow. 

Point 1: BA2H gene is so important in SA biosynthesis, but nobody has identified this gene sequence yet. The gene sequence in your manuscript might be wrong, which will be misleading readers. If you want to put the sequence of the BA2H in your manuscript, please do the enzyme activity in this manuscript.

Response 1: Thanks for your sincerely suggestion. After our carefully reconsideration and reconfirmation about your kindly reminder, we have removed the content of the BA2H gene in this manuscript including in the abstract and provided the revised figure 6, and Table S2.  Please see the revisions in the manuscript with highlighted in yellow.

This manuscript is a resubmission of an earlier submission. The following is a list of the peer review reports and author responses from that submission.

Round 1

Reviewer 1 Report

The manuscript is written carelessly. Authors must carefully check the text and correct all mistakes and formatting.

Figures are not in the right place in the text.

Why there are numbers in the abstract?

Such as

(1) Background:

References in the text must be corrected.

Why did you use both numbers and names of the authors? What does it mean?

Line 30

(Everts and Himmelstein 2015; Li et al. 2020; Zhu et al. 2020)[1].

Manuscript needs English corrections. Some sentences are incomplete.

For example,

lines 28-29

Watermelon (Citrullus lanatus) Fusarium wilt disease caused by Fusarium oxysporum f. sp. niveum (FON), which poses a serious threat to watermelon quality and yield

“is” is missing

line 70-71

“Thus, the results indicated that lower concentration of exogenous SA could effectively

decreased watermelon Fusarium wilt incidence which may help activate their disease

resistance.”

“SA could effectively decrease” OR “SA effectively decreased”

line 75-76

“Our data indicated that the SA was induced after FON pathogen

attack in watermelon roots which may activating watermelon plant resistance.”

“may be activating” OR “may activate”

line 93-95

“We therefore aimed to further explore the different gene expression pat-

terns related to SA pathways in watermelon roots which might responsible for Fusarium

wilt resistance under different cultivation systems.”

might be responsible

Please check through the text

Authors should check the abbreviations.

For example, FON (Fusarium oxysporum f. sp. Niveum) is mentioned twice in lines 28 and 52

In the Abstract SA abbreviation is not decoded.

Line 87-89

“And the results showed that there was a signifi-

cant enrichment of PAL enzyme activity (Fig 3D) but a decreasing of POD enzyme activ-

ity (Fig 3E) and MDA content (Fig 3F)”

Indicate, what is PAL, POD and MDA

line 59

two points

Watermelon Fusarium disease. .

It is not clear from the abstract and introduction if SA is good or bad for plants and FON and why. Please expand the text and add more information on SA and make it clear.

Line 53-54

Ren have detected that SA was secreted by watermelon roots, which can stimulate FON spore ger-

mination at low concentrations (Ren et al. 2016).

This sentence is not clearly written, it is hard to understand. Do you mean SA concentration or spore concentration?

Line 68-69

However, the 200 μM exogenous SA suppressed plant growth

and the SA content was significantly higher in the continuous monocropping system.

If you speak of exogenous SA, why the concentration was different in monocropping system?

You state that The incidence of watermelon Fusarium wilt in the rotated cropping system was significantly lower than that in the continuous cropping system, but the abundance of FON had no significant differences between these two systems.

Please explain the reasons, why plants could be healthier in rotated system, if the concentration of the pathogen was the same.

Line 85-86

“The fresh weight

of roots (Fig 3A) in R was nearly twice heavier compared with C group,”

please indicate what are R and C groups

This is the first mention of these groups, and not line 89-91

“Therefore, these results

showed that the plants under rotation (R) grew healthier than those under continuous

cropping (C).”

and not lines 101-102

“Samples, referred to as C (continuous watermelon monocropping)

and R (rotated with rape cropping).”

Fig. 1D

it can be seen that SA content increased at 3 and 7 dpi, but decreased at 5 dpi. Is there any explanation?

Did you use microscopy to study the effect of exogenous SA on fungal growth? Were there any in vitro experiments?

Where did you gen exogenous SA? (manufacturer)

Line 362-363

Did you measure decease severity? Which signs of infection did you take into account?

Author Response

Manuscript Submission ID plants-1433435 has been revised according to the reviewer and editorial suggestions. Reviewers’ constructive criticisms and suggestions are well received. (I) Generally, most of the suggested revisions involve minor changes that have been incorporated into the revision and marked up using the“Track Changes”function (see attached Final Revision for details). For example, we have rewritten the words in Fig. 5 B and Fig. 7, Supplement material 2. And we have deleted the sentence of "Phytophthora Database" in line 361-362. (II) Besides, we have have modified the repeated paragraphs and underwent extensive English revisions of this manuscript using the MDPI English editing services as you suggested, including the grammar, spelling, punctuation and phrasing checks, layout editing and iThenticate plagiarism check of this paper. For example, we have reconfirmed the style of our manuscript (especially figures and references). (III) The point by point, the details of the revisions to the manuscript and our responses to the referees’ comments are in the attachment file.

Reviewer 2 Report

The manuscript „Study on the role of Salicylic Acid in Watermelon Resistant Fusarium Wilt under Different Growth Conditions” presents an abundance of very interesting results. The findings seem to enrich the present knowledge about the role of SA in the defence against Fusarium wilt mechanisms of watermelon.

However I wrote “seem to”, because the paper is very confusing in many parts. First of all the MS needs extensive English editing prior to the next round of review. In the present form it is very hard to follow the Authors intentions, especially in the Discussion, and Materials and methods part. There were sentences which I could not comprehend despite several rounds of reading, and despite the fact that I understood each word separately.

The second serious flaw is the lack of any explanation and discussion of the fact that only one variety was used in this study. How the authors find it possible to conclude about all watermelons on the basis of one variety only? As it is not explained in the present version of the MS, I find the fact of using only one cultivar disqualifying for the design of this study. But if it was properly introduced and discussed I could be persuaded otherwise.

Third serious flaw is the description of materials and methods, especially RT qPCR experiments, but also partially RNAseq analysis. The authors seem to focus their description on irrelevant details, which aren’t usually mentioned in scientific papers (e.g. standard curves) and omit some of the important parts of the description. It is unclear what is the control and what objects are compared in parts of the MS (also in the results).

Also the style used in this MS is not according to the instructions for authors for this Journal (especially references), there should not be a separate paragraph with figures only, or a paragraph containing solely two line equation, with no comment.

More remarks, and more detailed comments the authors can find in the pdf which I attach to this review.

Author Response

(The authors gave the same response as above.)

Round 2

Reviewer 1 Report

1

line 19, «rotating rape cropping systems»

not clear what does it mean. I suppose it is called «oilseed rape rotation system» but you should check.

In the Abstract, you state that you study the effect of SA and cropping systems on plant resistance, but you only provide results of gene expression study.

Methods should correspond to the results.

2

to Response 11 «We have rewritten the sentences to expand the text and add more information on SA to plant health in the abstract and introduction. Please see line 17-18 and line 47-54.»

I don’t see much improvement. Introduction is still very small and studies on the role of SA in resistance to FON are barely mentioned. It is not rally clear, how SA is good for plants if, as you mention in the introduction, it can stimulate FON spore germination… Ren et al. 2016 observed the decrease in FON spore germination at high concentrations of SA, however you do not mention and do not discuss it.

How does SA affect FON while increasing plant resistance? What is the supposed mechanism? Why concentration is important and how does it affect plants? On line 115 you write that higher level of SA is harmful to the plants. Therefore SA appears to be bad for both plants and FON, and at low concentrations it is good for FON, is it not? Was there any concentration toxic to FON but normal for plants?

Also write more about genes you studied. Genes CIPALs and BA2H appear in the Abstract but there is no information about them in Introduction. What is known about them and other candidate genes in SA biosynthesis?

To “Response 15: Thanks for your kindly suggestion. We have rewritten the sentence with

explanation in line 80-84.”

Is there explanation to this? Please discuss it, maybe you can find some literature to find the reason why SA level dropped?

To Response 16: “Ren have reported that SA is secreted by

watermelon roots, which can stimulate FON spore germination (Ren et al. 2016). Therefore,

in this experiment, we have not used use microscopy to study the effect of exogenous SA on

fungal growth”

I do not see how it answers my questions about microscopy. Ren at all did not use it either. They also only studied spore germination and sporulation and did not reveal any mechanism of the SA effect on the fungi.

To Response 18: “we calculated the incidence of Fusarium wilt as while as the obvious symptoms appeared such as plant yellowing and wilting. And we added the explanations in line 408-409.”

You have written that the phenotypes of the watermelon seedling were compared and disease incidences at 7 dpi (7 days post inoculation) were recorded. And that these obvious symptoms were “such as plant yellowing and wilting”. Was there a control group that was not exposed to the treatment? Were there any symptoms in this group? These symptoms are common for many stress factors and can appear due to other reasons. Here could be more pathogens, especially in in monocropping systems.

on Fig. 1 B I can’t see decease incidence in different types of cultivation conditions (cropping systems. In which group was it measured, and why only in one group?

Author Response

Point 1: line 19, «rotating rape cropping systems» not clear what does it mean. I suppose it is called «oilseed rape rotation system» but you should check.

Response 1: Thanks for your kindly suggestion.

We have changed all the “rotating rape cropping systems” to “oilseed rape rotation system” in the manuscript including in line 19.

Point 2: In the Abstract, you state that you study the effect of SA and cropping systems on plant resistance, but you only provide results of gene expression study. Methods should correspond to the results.

Response 2: Thanks for your kindly suggestion. We have added the methods to study the effect of SA and cropping systems on plant resistance in the Abstract. Please see line 17.

Point 3: to Response 11 «We have rewritten the sentences to expand the text and add more information on SA to plant health in the abstract and introduction. Please see line 17-18 and line 47-54.»

I don’t see much improvement. Introduction is still very small and studies on the role of SA in resistance to FON are barely mentioned. It is not rally clear, how SA is good for plants if, as you mention in the introduction, it can stimulate FON spore germination… Ren et al. 2016 observed the decrease in FON spore germination at high concentrations of SA, however you do not mention and do not discuss it.

Response 3: Thanks for your kindly suggestion. We have added the information about the role of SA in resistance to FON and discuss it about how SA is good for plants in the introduction section. Please see line 39-41 and line 54-57.

Point 4: How does SA affect FON while increasing plant resistance? What is the supposed mechanism? Why concentration is important and how does it affect plants? On line 115 you write that higher level of SA is harmful to the plants. Therefore SA appears to be bad for both plants and FON, and at low concentrations it is good for FON, is it not? Was there any concentration toxic to FON but normal for plants?

Response 4: Thanks for your kindly suggestion. We have added the information about How does SA affect FON while increasing plant resistance? What is the supposed mechanism? Why concentration is important and how does it affect plants? Please see line 47-52. We did not find any reports about whether there is a certain concentration toxic to FON but normal for plants. But we added reference about that the exogenous SA concentrations inhibited Foc TR4 growth by Wang et al., to further explain the effect of SA on inhibit pathogen growth. Please see line 54-55. Moreover, this is a good suggestion for our present work about the effect of SA remodeling rhizosphere microbiota on inducing watermelon resistance against Fusarium wilt, which gives us another way to figure out whether the certain SA concentration influence the FON growth help watermelon recruiting the rhizosphere microbiota structure. And we are planning on publishing this work as soon as possible.

Point 5: Also write more about genes you studied. Genes CIPALs and BA2H appear in the Abstract but there is no information about them in Introduction. What is known about them and other candidate genes in SA biosynthesis?

Response 5: Thanks for your kindly suggestion. We have added the information of PALs and BAH genes in the introduction section. Please see line 41.

Point 6: To “Response 15: Thanks for your kindly suggestion. We have rewritten the sentence with explanation in line 80-84.”

Is there explanation to this? Please discuss it, maybe you can find some literature to find the reason why SA level dropped?

Response 6: Thanks for your kindly suggestion. We have added the explanation and discuss the reason why SA level dropped with references in line 80-82.

Point 7: To Response 16: “Ren have reported that SA is secreted by watermelon roots, which can stimulate FON spore germination (Ren et al. 2016). Therefore, in this experiment, we have not used use microscopy to study the effect of exogenous SA on fungal growth” 

I do not see how it answers my questions about microscopy. Ren at all did not use it either. They also only studied spore germination and sporulation and did not reveal any mechanism of the SA effect on the fungi.

Response 7: Thanks for your kindly suggestion. Sorry for my misunderstanding of your question earlier and we did not use microscopy to study the effect of exogenous SA on fungal growth in this experiment. However, this is a good advice for our present and further research work,which will help us further reveal the mechanism of the SA effect on FON.

Point 8: To Response 18: “we calculated the incidence of Fusarium wilt as while as the obvious symptoms appeared such as plant yellowing and wilting. And we added the explanations in line 408-409.”

You have written that the phenotypes of the watermelon seedling were compared and disease incidences at 7 dpi (7 days post inoculation) were recorded. And that these obvious symptoms were “such as plant yellowing and wilting”. Was there a control group that was not exposed to the treatment? Were there any symptoms in this group? These symptoms are common for many stress factors and can appear due to other reasons. Here could be more pathogens, especially in in monocropping systems.

Response 8: Thanks for your kindly suggestion. We have added some information about the phenotypes of watermelon seedling and roots after FON infection in other groups. Please see line 70-71 and line 73-74. Besides, we have added a subsection in methods (4.2.) section to emphasize our experiment about detection and incubation of FON in line 368 and added more explanations about how we measure the disease incidence in line 381.

Point 9: on Fig. 1 B I can’t see decease incidence in different types of cultivation conditions (cropping systems. In which group was it measured, and why only in one group?

Response 9: Thanks for your kindly suggestion. We have displayed the comparison result of disease incidence after five different concentrations of exogenous SA (0um, 10um, 50um, 100um, and 200um) to select the optimal SA concentration which leading to help watermelon plants growth after FON infection on Fig.1B. The decease incidence in different types of cultivation conditions is performed on Fig.2B. We have added more information in the title of Fig1 and Fig2 to distinguish the results about these two experiments. Please see line 84-85 and line 118.

Reviewer 2 Report

The manuscript has been corrected by the authors and was subjected to proper English editing. There are still some minor editing issues (e.g. hyphens in the middle of words that sould not be there), but I am sure that the editorial service will help with that.

However the issue of explaining the reason of using only one variety was not adressed properly by the authors, in my opinion.

The explanation that was given in the response letter was sufficient to me but the explanation used in the manuscript (only that it is the main cultivar grown in China) - not. The authors should use the reasons they described in the response to my previous remark in the discussion section.

Author Response

Point 1: The manuscript has been corrected by the authors and was subjected to proper English editing. There are still some minor editing issues (e.g. hyphens in the middle of words that should not be there), but I am sure that the editorial service will help with that.

Response 1: Thanks for your kindly suggestion.

I wonder these issues might be display error caused by the different version of Microsoft. We have revised some of those hyphens in the middle of words that should not be there. Besides, we have also contact with editor to make sure that the editorial service could doubt check with that.

Point 2: However the issue of explaining the reason of using only one variety was not adressed properly by the authors, in my opinion. The explanation that was given in the response letter was sufficient to me but the explanation used in the manuscript (only that it is the main cultivar grown in China) - not. The authors should use the reasons they described in the response to my previous remark in the discussion section.

Response 2: Thanks for your kindly suggestion.

We have added explanation the fact that why only one variety was used in the discussion section as you mentioned. “The most important reason why we used only one variety in this study is that the Zaojia 8424 of watermelon variety is the main cultivated variety in Chinese market and we have been done much field experiment research these years [3,6,7]”. Please see line 235-237 with highlight in the manuscript.

Round 3

Reviewer 1 Report

Dear author!

The Introduction is still very small. You barely added several new sentences.

These new sentences require language corrections.

1.

"Most researches have already

identified the vital role of salicylic acid (SA) in inducing plant defences to resistant

against pathogens"

The sentence is hard to understand. "inducing plant defences to resistant against " - the language is not correct

2.

line 41.

"ICS, PALs, and BAH genes in SA biosynthesis"

Please indicate the abbreviations of ICS, PAL, BAH and other genes. What is their exact role in SA biosynthesis? How do they mediate resistance?

Unfortunately I do not see much improvement here. You just listed the genes but did not describe them.

3.

"Moreover, other studies show that the application with exogenous SA with lower level en-

hanced resistance to Fusarium wilt"

The sentence is hard to understand. "application with exogenous SA with lower level" - the language is not correct

4.

"Ren reported that the certain concentrations of SA can stimulate FON spore

germination but good for plant growth [23]."

Which concentrations?

The sentence is hard to understand. "can stimulate FON spore

germination but good for plant growth" - the language is not correct

5.

"molecular mechanisms involved of SA in watermelon Fusarium wilt remain unknown."

The sentence is hard to understand. " involved of SA" - the language is not correct

6.

"The observations of wa-

termelon seedling results showed that there were more leaves and developed roots after

exogenous SA application at 100 μM as compared with the severe plant yellowing and

wilting in control group (0 μM, without SA treatment) (Figure 1A), and a significantly

lower disease incidence (Figure 1B)."

The sentence is hard to understand.

7.

"However, higher concentrations of SA suppressed

plant growth, such as the plant yellowing and wilting at 200 μM of exogenous SA ap-

plication (Figure 1A)."

the language is not correct. Did SA suppress plant growth or plant yellowing? It is not clear

8.

"Figure 1.B"

Once again, what was the disease incidence in plants that were not treated with SA?

9.

"The GO annotation analysis indi-

cated that there were 17 significantly differential expressed GO pathways compared

under different groups in our experiment (Figure 4D)."

" significantly differential expressed" - the language is not correct.

10.

I did not get an answer to this question: Was there a control group that was

not exposed to the treatment? Were there any symptoms in this group?